# Inherently Interpretable Multi-Label Classification Using Class-Specific Counterfactuals

**Susu Sun**[1]                                                                    SUSU.SUN@UNI-TUEBINGEN.DE

[1] *Cluster of Excellence – Machine Learning for Science, University of Tübingen, Germany*

**Stefano Woerner**[1]                                                      STEFANO.WOERNER@UNI-TUEBINGEN.DE

**Andreas Maier**[2]                                                                      ANDREAS.MAIER@FAU.DE

[2] *Pattern Recognition Lab, Friedrich-Alexander-Universität Erlangen-Nürnberg, Germany*

**Lisa M. Koch**[3,4]                                                             LISA.KOCH@UNI-TUEBINGEN.DE

[3] *Hertie Institute for Artificial Intelligence in Brain Health, University of Tübingen, Germany*

[4] *Institute of Ophthalmic Research, University of Tübingen, Germany*

**Christian F. Baumgartner**[1]                                   CHRISTIAN.BAUMGARTNER@UNI-TUEBINGEN.DE

**Editors:** Accepted for publication at MIDL 2023

## Abstract

Interpretability is essential for machine learning algorithms in high-stakes application fields such as medical image analysis. However, high-performing black-box neural networks do not provide explanations for their predictions, which can lead to mistrust and suboptimal human-ML collaboration. Post-hoc explanation techniques, which are widely used in practice, have been shown to suffer from severe conceptual problems. Furthermore, as we show in this paper, current explanation techniques do not perform adequately in the multi-label scenario, in which multiple medical findings may co-occur in a single image. We propose Attri-Net[1], an inherently interpretable model for multi-label classification. Attri-Net is a powerful classifier that provides transparent, trustworthy, and human-understandable explanations. The model first generates class-specific attribution maps based on counterfactuals to identify which image regions correspond to certain medical findings. Then a simple logistic regression classifier is used to make predictions based solely on these attribution maps. We compare Attri-Net to five post-hoc explanation techniques and one inherently interpretable classifier on three chest X-ray datasets. We find that Attri-Net produces high-quality multi-label explanations consistent with clinical knowledge and has comparable classification performance to state-of-the-art classification models.

**Keywords:** Interpretable Machine Learning, Visual Feature Attribution, Multi-label Classification.

## 1. Introduction

The clinical adoption of machine learning (ML) technology is hindered by the black-box nature of deep learning models. Their inscrutability may lead to a lack of trust (Dietvorst et al., 2015), or blind trust among clinicians (Tschandl et al., 2020; Gaube et al., 2021), and may result in ethical as well as legal problems (Grote and Berens, 2020). Therefore, transparency has been identified as one of the key properties for deploying machine learning technology in high-stakes application areas such as medicine (Rudin, 2019).

---

1. The code for Attri-Net is available at https://github.com/ss-sun/Attri-Net

The most commonly used category of techniques for understanding the decision mechanisms of ML models are *post-hoc* methods which apply a heuristic to a trained model trying to understand the decision mechanism retrospectively after the prediction is made. Gradient-based techniques such as Guided Backpropagation (Springenberg et al., 2014) perform local function approximation of the black-box model by differentiating the prediction with respect to the input pixels. The faithfulness of such methods to the decision mechanisms has recently been put into question by Adebayo et al. (2018) and Arun et al. (2021) who showed that explanations remain unchanged despite randomisation of network weights. Perturbation-based methods such as LIME (Ribeiro et al., 2016), or SHAP (Lundberg and Lee, 2017) also approximate the local decision function. These methods cannot currently produce explanations at the pixel level and are computationally demanding. Another line of work including Class Activation Mappings (CAM) (Zhou et al., 2016) and GradCAM (Selvaraju et al., 2017) attempts to construct neural network architectures from which the decision mechanism can be directly inferred. However, these techniques are limited by the spatial resolution of their explanations and do not explain the reasoning mechanism on a pixel-level. BagNet (Brendel and Bethge, 2019) addresses this issue by severely restricting the global receptive field of the network which can negatively affect classification performance. Placing attention modules at different depths throughout the network can also provide a measure of interpretability to individual feature maps (Schlemper et al., 2019; Yan et al., 2019). A category of approaches highly related to our proposed method are counterfactual explanations which either try to answer the question "What would the image look like if it belonged to a different class?" (Schutte et al., 2021; Joshi et al., 2018), or exaggerate the features of the predicted class (Cohen et al., 2021; Singla et al., 2019). Other approaches in this category derive classifications from an intermediate representation of the counterfactual generator (Bass et al., 2020; Cetin et al., 2022). We also note that some techniques attempt to generate counterfactuals without the aim of explaining a classifier (Baumgartner et al., 2018; Nemirovsky et al., 2020).

While *post-hoc* explanations may appear reasonable, there is no guarantee that they explain what the classifier actually does, and there is, in fact, growing evidence that they are not faithful to the actual decision mechanism (Adebayo et al., 2018; Han et al., 2022; White and Garcez, 2019). In contrast, *inherently interpretable* methods use prediction systems for which the decision mechanism is directly revealed to the user. These models are by definition faithful to the decision mechanism because the explanation *is* the decision mechanism. Prior work includes methods in which the final predictions are directly based on human-interpretable concepts (Alvarez Melis and Jaakkola, 2018; Chen et al., 2020; Koh et al., 2020), prototypical representations of classes (Chen et al., 2019; Barnett et al., 2021), or direct attribution to image patches (Javed et al., 2022). The recently proposed Convolutional Dynamic Alignment Networks (CoDA-Nets) (Bohle et al., 2021) is, to our knowledge, the only existing model providing inherently interpretable visual explanations on the pixel-level. The method expresses network weights as a function of the input image in a way that allows them to formulate the networks' decision for a specific input image as a linear classifier. We note that there are to our knowledge no inherently interpretable methods based on counterfactual explanations.

The majority of visual explanation techniques were developed for binary or multi-class problems. Many clinical tasks, however, are multi-label problems, where multiple classes

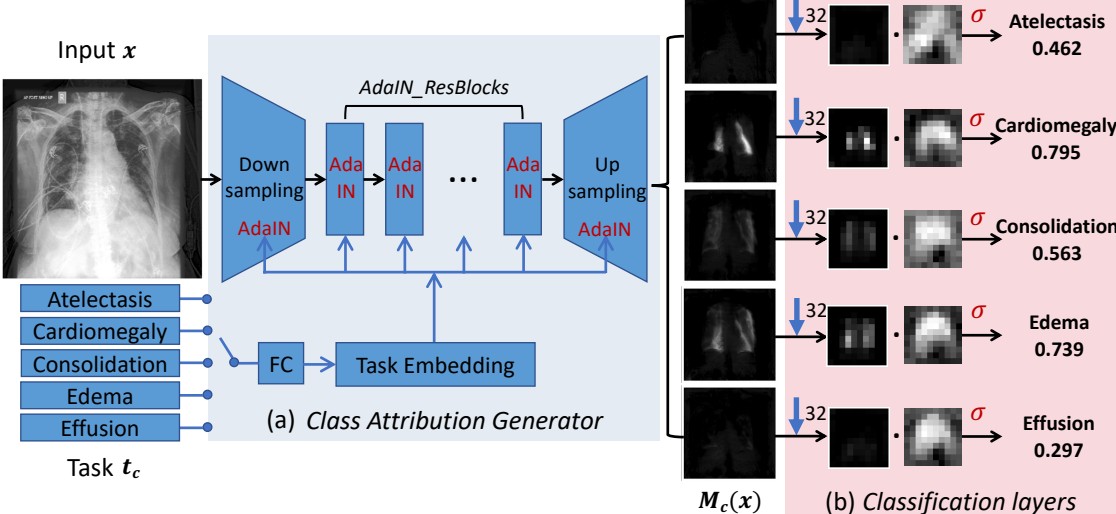

Figure 1: Attri-Net Framework. Given an input image and a task, our visual feature attribution generator (a) produces counterfactual attribution maps. Based on these maps a logistic regression classifier (b) produces the final prediction for each class.

can apply simultaneously. For example, in chest X-ray diagnosis, which we study in this work, an image often contains multiple findings. In our experiments, we found that existing visual explanation techniques are not well-suited to this important type of decision problem. In particular, the explanations are not specific to the class and tend to highlight similar regions for all classes, in some cases even when the class is not present in the image.

In this paper, we propose Attri-Net, an inherently interpretable visual explanation technique designed specifically for the multi-label scenario. Our model predicts class-specific counterfactual attribution maps as intermediate representations. The attribution maps, which are conceptually based on the visual feature attribution GAN (VA-GAN) approach introduced by Baumgartner et al. (2018), represent residual images that contain all existing evidence of a class in an input image. Attri-Net then uses these class-specific attribution maps as input features in a final linear classification layer.

We evaluate Attri-Net on three widely used chest X-ray datasets and demonstrate that the method produces high-quality inherently interpretable explanations with a high class sensitivity while retaining classification performance comparable to state-of-the-art models.

## 2. Methods

In this paper, we address the multi-label classification scenario with $C$ classes, where each class $c$ with label $y_c \in \{0, 1\}$ can independently occur in an image, i.e. multiple co-existing medical findings are possible. In the following, we first introduce our method for generating counterfactual class attribution maps for each class $c$ (see Fig. 1a). Then, we show how a logistic regression classifier is used to obtain the final predictions based on those attribution maps (see Fig. 1b). Lastly, we explain how these two components are trained end-to-end in our proposed Attri-Net framework.

## 2.1. Counterfactual class attribution map generation

The core of our method is an image-to-image network $M_c(\mathbf{x}) : \mathbb{R}^{h \times w} \mapsto \mathbb{R}^{h \times w}$ which generates residual counterfactual class attribution maps for an input image $\mathbf{x}$. Intuitively, the output of $M_c$ represents how each pixel in the input should change in order to remove the effect of class $c$ from the image. Like Baumgartner et al. (2018), we learn an additive mapping $M_c$ that makes the output image appear to come from the opposite class, that is

$$\hat{\mathbf{x}} = \mathbf{x} + M_c(\mathbf{x}),$$

such that the generated counterfactual image $\hat{\mathbf{x}}$ is indistinguishable from images sampled from the distribution $p(\mathbf{x}|y_c = 0)$ of real images *not* containing class $c$. To ensure the correct behavior of $M_c$, we simultaneously train a class-specific discriminator network $D_c$ to distinguish between real and fake images with $y_c = 0$. Specifically, we use the Wasserstein GAN loss (Arjovsky et al., 2017; Baumgartner et al., 2018). Details on the optimisation of $D_c$ are given in Appendix B.1. Given a discriminator function $D_c$ we can write the following adversarial loss term ensuring that $\hat{\mathbf{x}}$ is a realistic counterfactual not containing class $c$ and, by extension, that $M_c$ outputs realistic residual class attribution maps:

$$\mathcal{L}_{\text{adv}}^{(c)} = \mathop{\mathbb{E}}_{\mathbf{x} \sim p(\mathbf{x}|y_c=1)} [-D_c(\mathbf{x} + M_c(\mathbf{x}))]. \tag{1}$$

Some examples of generated counterfactuals $\hat{\mathbf{x}}$ are shown in Appendix A.1.

To discourage the network from attributing superfluous pixels not belonging to a given class, we additionally encourage the class attribution maps to be sparse using an $L_1$ regularization term similar to Baumgartner et al. (2018). To further encourage the generator to produce smaller effects when the class is present in an image than when it is not present, we divide the loss into two differently weighted terms with a larger weight $\alpha_0$ for class-negative, and a smaller weight $\alpha_1$ class-positive examples, i.e.,

$$\mathcal{L}_{\text{reg}}^{(c)} = \alpha_0 \mathop{\mathbb{E}}_{\mathbf{x} \sim p(\mathbf{x}|y_c=0)} [\|M_c(\mathbf{x})\|_1] + \alpha_1 \mathop{\mathbb{E}}_{\mathbf{x} \sim p(\mathbf{x}|y_c=1)} [\|M_c(\mathbf{x})\|_1]. \tag{2}$$

We use $\alpha_0 = 2, \alpha_1 = 1$ for all experiments in this paper.

The functions $M_c$, and $D_c$ are implemented as neural networks building on the StarGAN architecture (Choi et al., 2018) which produced superior results to alternative options we explored such as the original VA-GAN architecture. Although it is feasible to design a network $M$ to produce class attribution maps for all labels as multiple output channels in a single forward pass, preliminary experiments revealed inadequate class attribution in the multi-label scenario. Instead, we build on the recently proposed task switching network (Sun et al., 2021) where adaptive instance norm (AdaIN) layers are used to switch between related tasks. In our work, tasks correspond to the generation of attribution maps for different classes. Each task is represented as a task vector $\mathbf{t}_c$ which is a one-hot encoding spatially upsampled by a factor of 20 as in (Sun et al., 2021). This encoding is then converted into a task embedding via a small fully connected network and fed to AdaIN layers which are placed throughout the network (as shown in Fig. 1a). The AdaIN layers then toggle the behaviour of the network. To combine this paradigm with the StarGAN architecture, we replaced all instance normalization layers of the original generator and discriminator

networks with AdaIN layers. The architecture is described in greater detail in Appendix B.2. The mask generator and discriminator can now be expressed as $M_c(\mathbf{x}) = M(\mathbf{x}, \mathbf{t}_c)$, and $D_c(\mathbf{x}) = D(\mathbf{x}, \mathbf{t}_c)$, respectively. The class attribution maps for all labels can be obtained by repeated forward passes through $M$ while iterating through the $\mathbf{t}_c$ vectors of all classes.

### 2.2. Classification using a logistic regression classifier

Given a class-specific counterfactual attribution map obtained using $M(\mathbf{x}, \mathbf{t}_c)$, we want to predict the presence of class $c$ in an image. To achieve this, the respective attribution map is downsampled and used as input to a logistic regression classifier. That is,

$$p(y_c|\mathbf{x}) = \sigma\Big(\sum_{i,j} w_{ij}^{(c)} \cdot S_\gamma(M(\mathbf{x}, \mathbf{t}_c))_{ij}\Big), \tag{3}$$

where $S_\gamma$ is a 2D average pooling operator that downsamples by a factor of $\gamma$, $w_{ij}^{(c)}$ denotes the weights associated with each pixel of the down-sampled attribution map for class $c$, and $\sigma$ is the sigmoid function. In preliminary experiments, we found $\gamma = 32$ to perform robustly and we use this value for all experiments.

The classifier is trained using a standard binary classification loss $\mathcal{L}_{\text{cls}}^{(c)}$, i.e. binary cross entropy loss for each class. Note that, since our framework is trained end-to-end, $M$ also receives gradients from that loss and is thereby encouraged to create class attribution maps that are linearly classifiable.

To further encourage the attribution maps to be discriminative for positive and negative examples of each class, we apply the center loss proposed by Wen et al. (2016), which has been shown to lead to more discriminative feature representations. Extending the idea, here, we define class centers $\mathbf{v}_{y_c=0}, \mathbf{v}_{y_c=1} \in \mathbb{R}^{h \times w}$ which are learnable and converge to prototypical representations of attribution maps corresponding to positive and negative instances of each class $c$. The center loss draws the class attribution maps closer to their respective class centers, resulting in a more clustered feature space where positive and negative samples are better linearly separable. The overall center loss can be written as

$$\mathcal{L}_{\text{ctr}}^{(c)} = \frac{1}{2}\left(\underset{\mathbf{x} \sim p(\mathbf{x}|y_c=0)}{\mathbb{E}}\left[\|M(\mathbf{x}, \mathbf{t}_c) - \mathbf{v}_{y_c=0}\|_2^2\right] + \underset{\mathbf{x} \sim p(\mathbf{x}|y_c=1)}{\mathbb{E}}\left[\|M(\mathbf{x}, \mathbf{t}_c) - \mathbf{v}_{y_c=1}\|_2^2\right]\right). \tag{4}$$

The class center images are updated for each mini-batch in a separate gradient update interleaved with the updates of the network parameters as described by Wen et al. (2016). The final class center images as well as the logistic regression weights $w_{ij}^{(c)}$ may be used to further interpret the model's behaviour on a global level. Examples of both are shown in Appendix A.2. However, we leave the exploration of global interpretability to future work.

### 2.3. Training

Our Attri-Net framework can be trained end-to-end with four loss terms enforcing our essential requirements: Firstly, the attribution map should preserve sufficient class relevant information such that a satisfactory classification result can be obtained. Secondly, the

Table 1: Classification performance measured by area under the ROC curve (AUC).

| Model | CheXpert | ChestX-ray8 | VindrCXR |
|---|---|---|---|
| ResNet50 (Azizi et al., 2021) | 0.7687 | - | - |
| SimCLR (Azizi et al., 2021) | 0.7702 | - | - |
| LSE (Ye et al., 2020) | - | 0.7554 | - |
| ChestNet (Ye et al., 2020) | - | 0.7896 | - |
| ResNet50 | 0.7727 | 0.7445 | 0.8986 |
| CoDA-Nets | 0.7659 | 0.7727 | 0.9322 |
| ours | 0.7405 | 0.7762 | 0.9405 |

Table 2: Comparison of class sensitivity scores.

| Model | CheXpert | ChestX-ray8 | VindrCXR |
|---|---|---|---|
| ResNet + GB | 0.3183 | 0.3028 | 0.1727 |
| ResNet + GCam | 0.1434 | 0.1570 | 0.1931 |
| ResNet + LIME | 0.2347 | 0.2609 | 0.2422 |
| ResNet + SHAP | 0.4745 | 0.4122 | 0.3714 |
| ResNet + Gifsplan. | 0.2748 | 0.5817 | 0.4396 |
| CoDA-Nets | 0.3576 | 0.4138 | 0.4464 |
| ours | **0.4880** | **0.6160** | **0.5509** |

attribution maps should be human-interpretable. The overall training objective for the class attribution generator $M$ with weight parameters $\varphi$ is given by

$$\min_{\varphi} \sum_c \lambda_{\mathrm{cls}} \mathcal{L}_{\mathrm{cls}}^{(c)} + \lambda_{\mathrm{adv}} \mathcal{L}_{\mathrm{adv}}^{(c)} + \lambda_{\mathrm{reg}} \mathcal{L}_{\mathrm{reg}}^{(c)} + \lambda_{\mathrm{ctr}} \mathcal{L}_{\mathrm{ctr}}^{(c)}, \tag{5}$$

where we use the hyperparameters $\lambda_*$ to balance the losses. We chose $\lambda_{\mathrm{cls}} = 100$, $\lambda_{\mathrm{adv}} = 1$, $\lambda_{\mathrm{reg}} = 100$, $\lambda_{\mathrm{ctr}} = 0.01$ for our experiments. An ablation study on the effect of the different losses can be found in Appendix A.3. During training, we repeatedly iterate through the different classes $c$ and, for each, sample two mini-batches, one containing positive examples of the current class and the other negative examples. We iteratively update $M$, $D$ and classifiers, and additionally train discriminator $D$ and classifiers more steps to ensure good feedback to mask generator $M$. We use the ADAM optimizer (Kingma and Ba, 2014) with a learning rate of $10^{-4}$ and a batch size of 4 to optimize our model. Furthermore, following Wen et al. (2016), we use stochastic gradient descent for updating the center loss parameters. Training converges within 72 hours on an Nvidia V100 GPU. After training we select the decision threshold which maximises the Youden-index (sensitivity + specificity - 1) for each class on the validation set. We also perform this step for the baseline methods.

## 3. Experiments and Results

**Data.** We evaluated our proposed Attri-Net on the three widely used chest X-ray datasets CheXpert (Irvin et al., 2019), ChestX-ray8 (Wang et al., 2017), and VinDrCXR (Nguyen et al., 2020). Following (Irvin et al., 2019) and (Azizi et al., 2021) for the CheXpert and ChestX-ray8 datasets we used the classes "Atelectasis", "Cardiomegaly", "Consolidation",

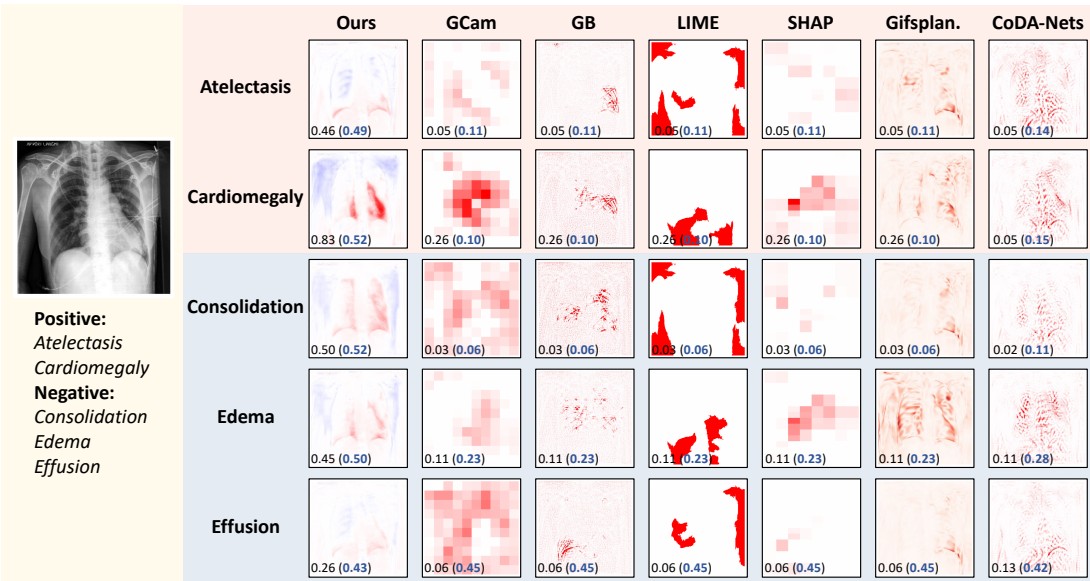

Figure 2: Visual comparison of explanations for an example image from the CheXpert dataset. Predicted class probabilities are indicated in the lower left corner of each attribution map with the respective decision threshold in parentheses.

"Edema", and "Pleural Effusion". For the VinDr-CXR dataset, we selected the five pathologies with the highest number of samples, which were "Aortic enlargement", "Cardiomegaly", "Pulmonary fibrosis", "Pleural thickening", and "Pleural effusion". We split all datasets into a training (80%), testing (10%) and validation (10%) fold. Since the test set of Chexpert was not publicly available and the official validation set was small, we adopted the method used in (Azizi et al., 2021) to split the official train set into train, validation, and test sets.

**Classification performance.** To assess the classification performance, we compared our model with the state-of-the-art inherently interpretable model CoDA-Nets (Bohle et al., 2021) as well as a standard black-box ResNet50 model. We also report the results of Azizi et al. (2021) and Ye et al. (2020) on CheXpert and ChestX-ray8, respectively. Attri-Net overall performed comparable to the state-of-the-art (see Tab. 1), with an area under the ROC curve that was slightly lower on CheXpert, similar to other methods on ChestX-ray8, and slightly better on VindrCXR.

**Interpretability.** Khakzar et al. (2021) argue that if different areas of an image are responsible for predicting different classes, then also the explanations should be different. They coin this property "class sensitivity". In the context of multi-label classification, the explanation for an image containing a class should have higher attribution than an image where the class is absent. We measured class sensitivity following Bohle et al. (2021) and created a series of $2 \times 2$ grids of explanations, where each grid contained only one positive example of a given class (see Appendix A.4 for example grids). We then represented class sensitivity by the sum of attributions in the positive example divided by the sum of all attributions in the grid. The optimal scenario where only the disease positive map contains

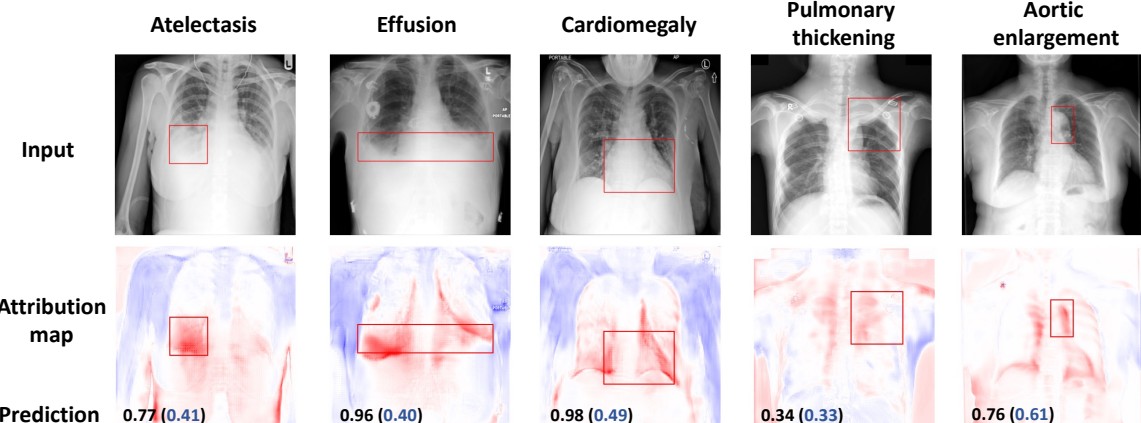

Figure 3: Attribution maps for samples from ChestX-ray8 and VindrCXR with GT bounding boxes. Decision thresholds are given in parentheses.

any attributions, and disease negative attribution maps are blank, yields a sensitivity of 1. We computed the average sensitivity over 200 grids for each class $c$.

Our method led to a substantially and consistently higher class sensitivity than the inherently interpretable baseline, CoDA-Nets, across all datasets (see Tab. 2). For the black-box ResNet, we compared five post-hoc explanations techniques, i.e. Guided Back-propagation (Springenberg et al., 2014), GradCAM (Selvaraju et al., 2017), LIME (Ribeiro et al., 2016), SHAP (Lundberg and Lee, 2017) and the recently proposed Gifsplanation (Cohen et al., 2021). The post-hoc methods varied considerably with SHAP and Gifsplanation performing comparably to CoDA-Nets, but substantially worse than our Attri-Net.

Qualitative examination of example explanations supported these results. Our proposed Attri-Net produced class attribution maps that clearly highlight the parts of the underlying anatomy that support the respective classes (see Fig. 2 for a representative example from the CheXpert dataset). Moreover, the attributions for different classes were clearly distinct from each other, each one focusing on different anatomical areas. Examples from the ChestX-ray8 and VinDr-CXR datasets can be found in Appendix A.5. In contrast, the inherently interpretable baseline, CoDA-Nets, produced visually similar attributions for all classes (rightmost column in Fig. 2). We further observed that the baseline techniques were mostly not useful for identifying which parts of the anatomy contributed to a prediction. While Guided Backpropagation qualitatively provided the most useful explanations of the baselines, its attributions were very noisy as is typical for this technique. We further examined Attri-Net explanations on example images of each class where pathology bounding boxes were available (Fig. 3). Attri-Net generally highlighted regions associated with the respective pathologies, with particularly sensitive attribution maps when the final prediction was highly confident (i.e. the examples with atelectasis, effusion, and cardiomegaly). We also observed some relatively strong attributions in regions outside the bounding boxes. As our class attribution maps were based on counterfactuals that were designed to realistically remove all effects of a pathology, we hypothesise they may have uncovered additional effects correlated with the classes which were not part of the clinical grading protocol.

## 4. Discussion and Conclusion

We proposed Attri-Net, a novel inherently interpretable multi-label classifier and showed that it produces high-quality explanations substantially outperforming all baselines in terms of class sensitivity while retaining classification performance comparable to state-of-the-art black-box models. Explanations of the black-box model were highly dependent on the post-hoc technique, and fundamentally differed from each other even on the same image. This erodes trust in their capacity to provide necessary transparency in high-stakes applications and shows the need for inherently interpretable models such as ours, where the predictions are formed directly and linearly from visually interpretable class attribution maps.

The qualitative and quantitative assessments in this paper suggest that our method provides useful explanations, but there remain important avenues for future work. We believe a crucial step towards clinical impact is the evaluation of interpretable models in actual human-ML collaboration setting to test their usefulness with clinically relevant endpoints.

## Acknowledgments

Funded by the Deutsche Forschungsgemeinschaft (DFG, German Research Foundation) under Germany's Excellence Strategy – EXC number 2064/1 – Project number 390727645. The authors acknowledge support of the Carl Zeiss Foundation in the project "Certification and Foundations of Safe Machine Learning Systems in Healthcare" and the Hertie Foundation. The authors thank the International Max Planck Research School for Intelligent Systems (IMPRS-IS) for supporting Susu Sun, Stefano Woerner, Lisa M. Koch, and Christian F. Baumgartner.

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

# Appendix A. Additional evaluations

## A.1. Examples of counterfactual generations

Examples of counterfactual images obtained by adding the class-specific visual attribution map to the input image, i.e. $\hat{\mathbf{x}} = \mathbf{x} + M_c(\mathbf{x})$, are shown in Fig. 4.

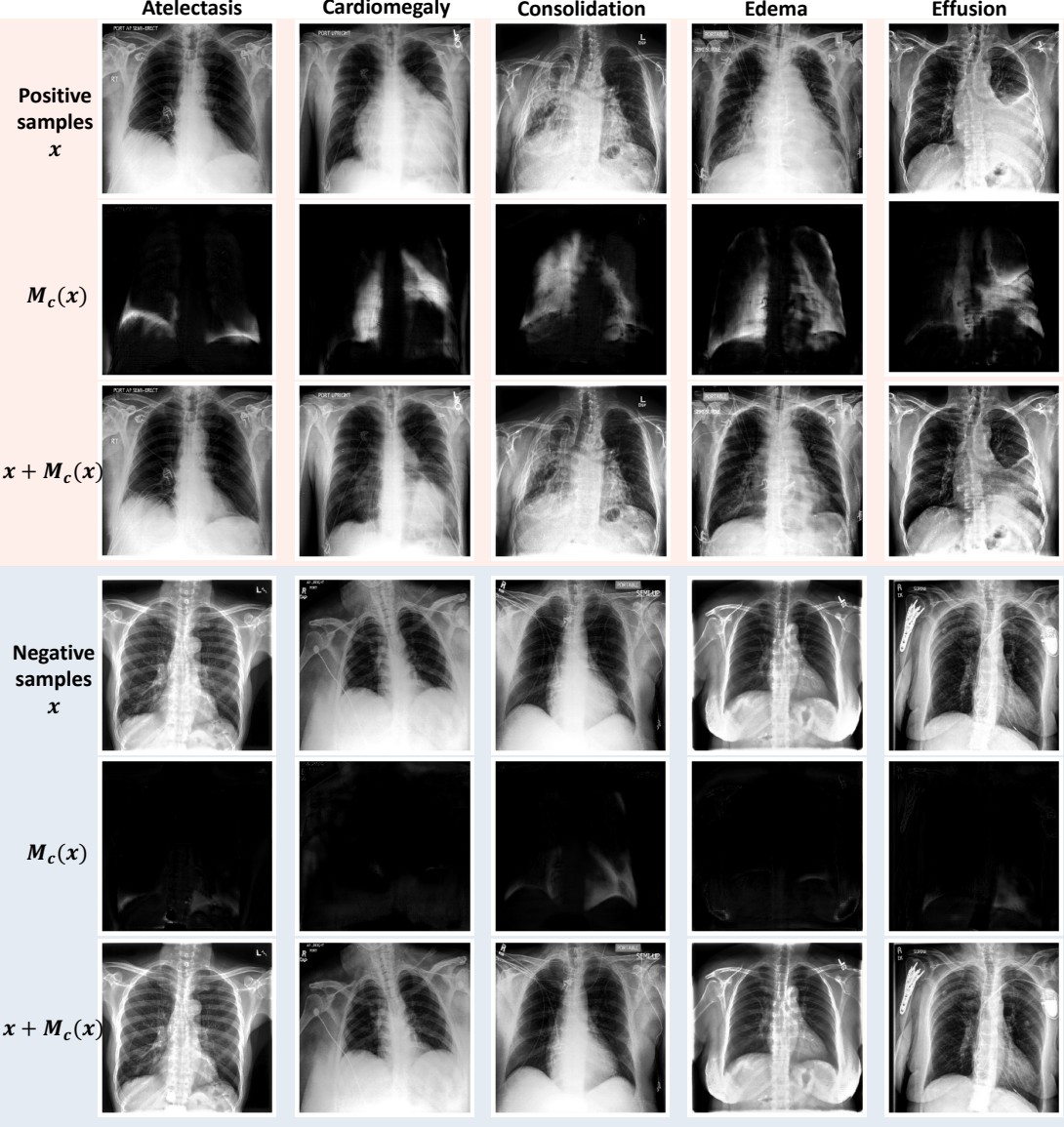

Figure 4: Counterfactual image generation. The examples in the top group of rows show input images containing evidence for different classes $c$. For those, the evidence for class $c$ is removed by adding $M_c(\mathbf{x})$. The bottom group of rows contains images with no evidence for class $c$. Those images remain mostly unchanged by adding the output of $M_c(\mathbf{x})$.

## A.2. Global interpretability

A distinction is often made between local explanations, which explain the prediction for a specific input image, and *global* explanations, which explain the decision mechanisms of the ML algorithm as a whole (i.e. for all input images). While the primary focus of our paper was on local interpretability, we may gain some global insights about the decision mechanism of the classifier through interpretation of the positive and negative class centers introduced in Section 2.2, as well as the class specific weights of the logistic regression classifier. The class centers capture some prototypical aspects of the respective classes, while the classifier weights can tell us which areas of the images the classifier is paying attention to for each class. Fig. 5 shows the class centers for five diseases and the weights of the corresponding classifiers trained on the ChestX-ray8 dataset.

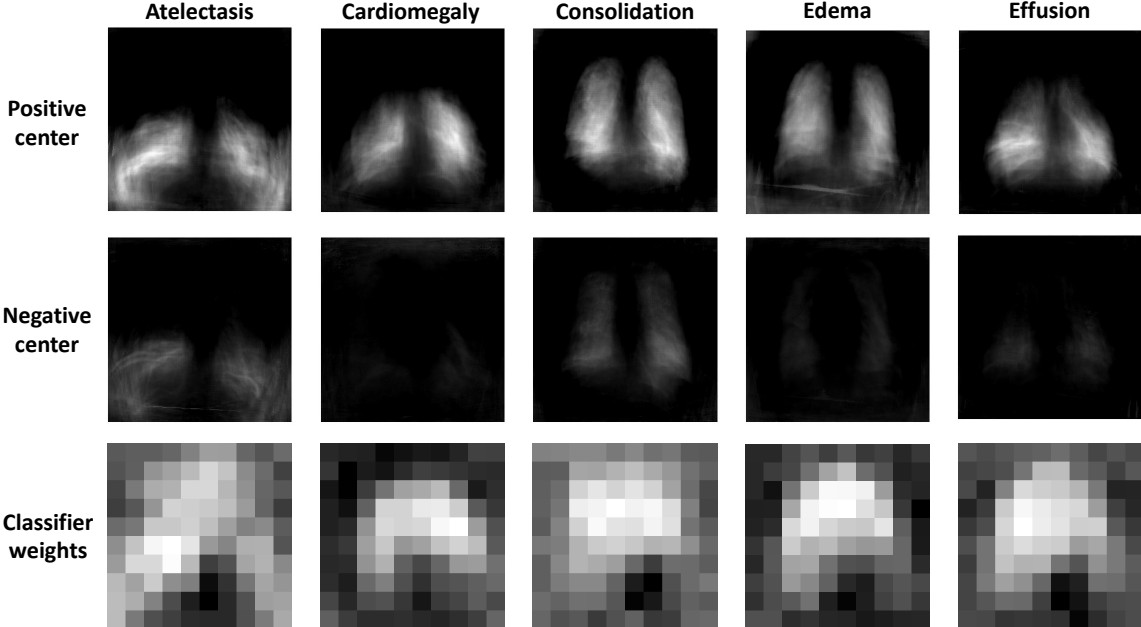

Figure 5: Attribution centers of disease and corresponding classifiers' weights provide a global explanation of Attri-Net.

## A.3. Ablation study of the loss terms

An ablation study on the effects of the losses used for training Attri-Net can be found in Tab. 3. Example attributions for all combinations for an image from the ChestX-ray8 dataset are shown in Fig. 6.

Table 3: Ablation study on the four losses. Evaluated on the Vindr-CXR dataset.

| Model | Loss terms | Classification AUCs | Class sensitivity |
|---|---|---|---|
| Attri-Net$_{\mathrm{cls}}$ | $\mathcal{L}_{\mathrm{cls}}$ | 0.9339 | 0.2516 |
| Attri-Net$_{\mathrm{cls\_adv}}$ | $\mathcal{L}_{\mathrm{cls}} + \mathcal{L}_{\mathrm{adv}}$ | **0.9444** | 0.1602 |
| Attri-Net$_{\mathrm{cls\_adv\_reg}}$ | $\mathcal{L}_{\mathrm{cls}} + \mathcal{L}_{\mathrm{adv}} + \mathcal{L}_{\mathrm{reg}}$ | 0.9397 | 0.5259 |
| Attri-Net$_{\mathrm{all}}$ | $\mathcal{L}_{\mathrm{cls}} + \mathcal{L}_{\mathrm{adv}} + \mathcal{L}_{\mathrm{reg}} + \mathcal{L}_{\mathrm{ctr}}$ | 0.9405 | **0.5509** |

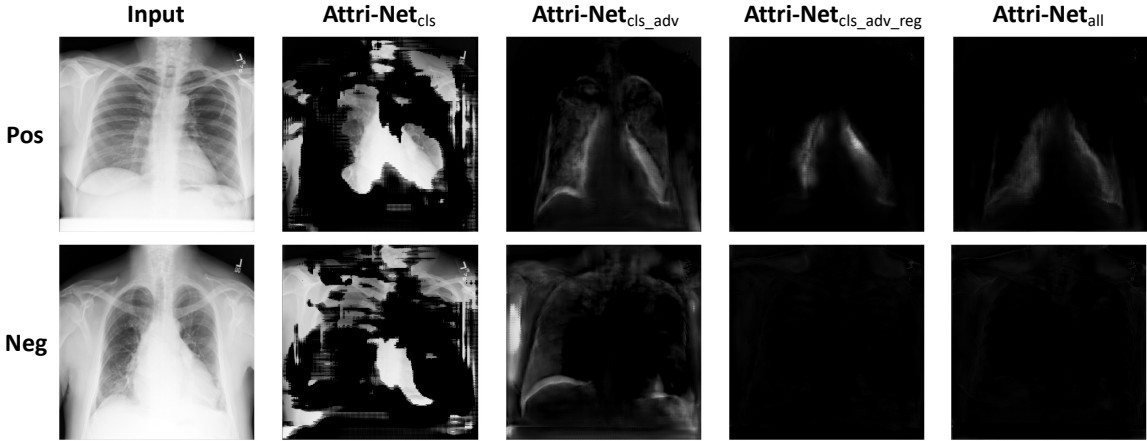

Figure 6: Examples of attribution maps obtained for positive and negative samples of the disease cardiomegaly on ChestX-ray8 for different subsets of our losses.

### A.4. Class sensitivity image grids

The class sensitivity evaluation in Section 3 is based on class sensitivity grids as proposed by Bohle et al. (2021). In Fig. 7, Fig. 8, and Fig. 9 we show examples of such grids for all studied classes on the CheXpert dataset, for Attri-Net, CoDA-Nets and Gifsplanation, respectively.

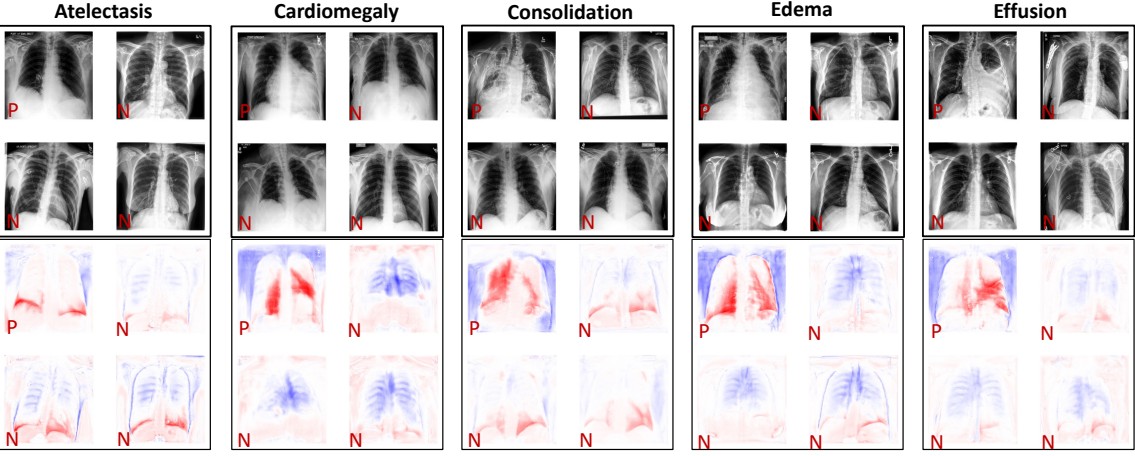

Figure 7: Class sensitivity image grids obtained using Attri-Net. The first row shows image grids, the second row shows the respective attribution maps. P and N denote class-positive and class-negative examples, respectively.

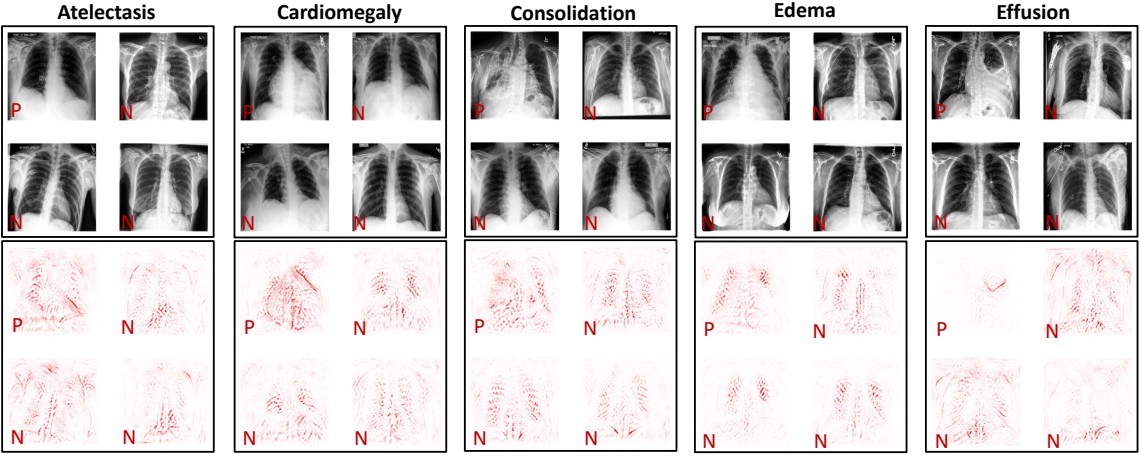

Figure 8: Class sensitivity image grids obtained using CoDA-Nets. The first row shows image grids, the second row shows the respective attribution maps. P and N denote class-positive and class-negative examples, respectively.

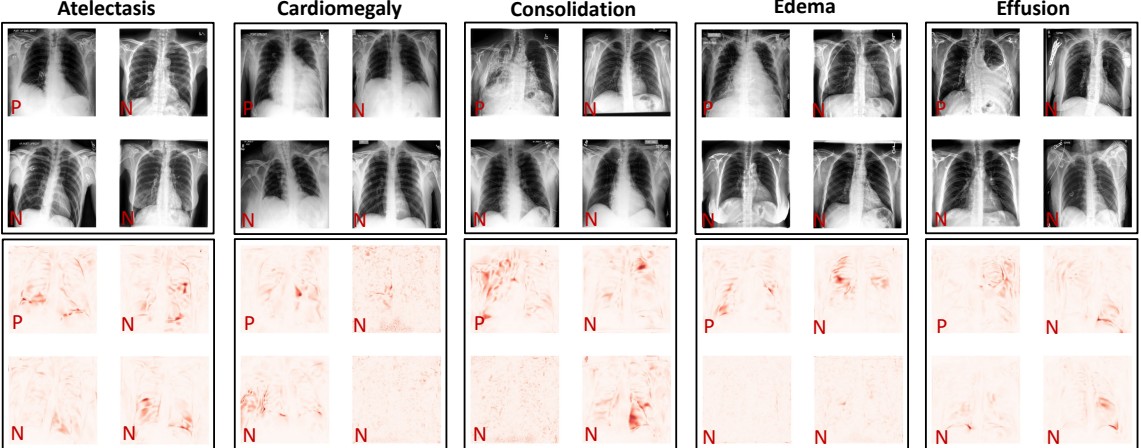

Figure 9: Class sensitivity image grids obtained using Gifsplanation. The first row shows image grids, and the second row shows the respective attribution maps. P and N denote class-positive and class-negative examples, respectively.

## A.5. Example explanations for ChestX-ray8 and Vindr-CXR

Fig. 10 and Fig. 11 contain additional examples of visual attributions using all compared methods derived from the ChestX-ray8 and Vindr-CXR datasets, respectively.

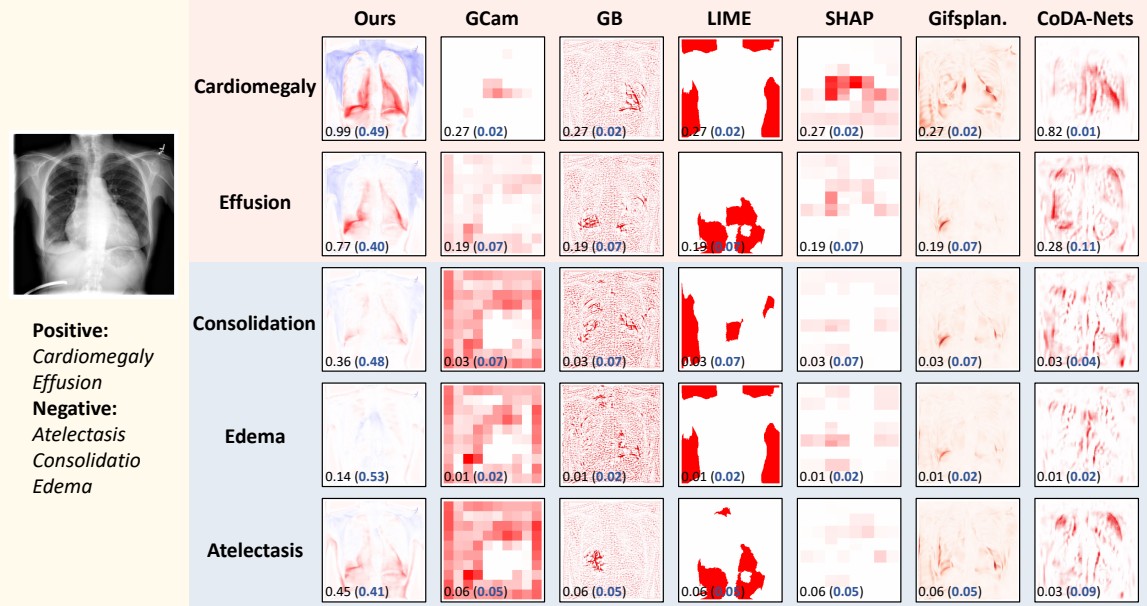

Figure 10: Explanations for an example image from the ChestX-ray8 dataset.

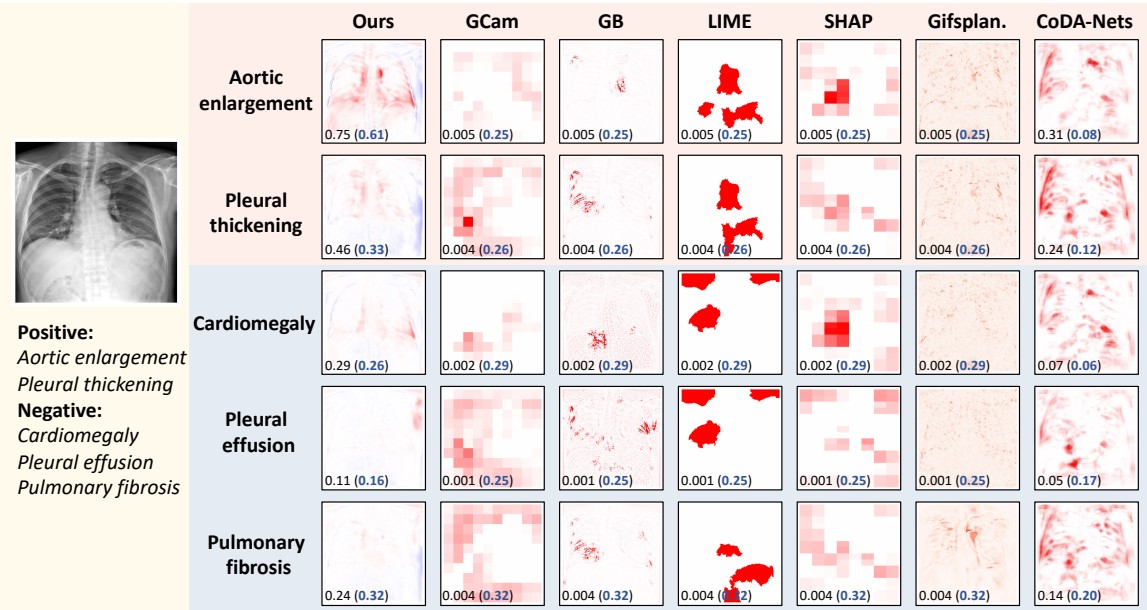

Figure 11: Explanations for an example image from the VindrCXR dataset.

## Appendix B. Additional training details

### B.1. Discriminator training

The Attri-Net framework requires training a discriminator function $D$ in parallel to the class attribution generator $M$. The weight parameters $\theta$ of the discriminator are computed in separate gradient update steps using the Wasserstein GAN (Arjovsky et al., 2017) objective. The full discriminator optimisation objective is then given by

$$\min_{\theta} \sum_{c} \mathbb{E}_{\mathbf{x} \sim p(\mathbf{x}|y_c=0)} [D_c(\mathbf{x}|\theta)] + \mathbb{E}_{\mathbf{x} \sim p(\mathbf{x}|y_c=1)} [D_c(\mathbf{x} + M_c(\mathbf{x})|\theta)],$$

where we omitted the gradient penalty loss which ensures the discriminator fulfills the Lipschitz-1 constraint dictated by the Wasserstein GAN objective (Gulrajani et al., 2017).

### B.2. Network architecture

The network architecture of the attribution map generator and the discriminator of the Attri-Net framework are shown in Tab. 4 and Tab. 5, respectively. L refers to the length of input/output features, N is the number of output channels, and K is the kernel size.

Table 4: Attri-Net class attribution generator network architecture.

| Layers | Input → Output | Layer information |
|---|---|---|
| **Task embedding layer** | Task code $\mathbf{t}_c$ → Task embedding $\mathbf{t}'_c$ | $8 \times$ FC(L100,L100) |
| **Down-sampling** | (Input image $\mathbf{x}$, $\mathbf{t}'_c$) → $\mathbf{out}_{\mathrm{down}}$ | Ada_Conv: CONV(N64, K7x7), AdaIN, ReLU 
 Ada_Conv: CONV(N128, K4x4), AdaIN, ReLU 
 Ada_Conv: CONV(N256, K4x4), AdaIN, ReLU |
| **Bottlenecks** | ($\mathbf{out}_{\mathrm{down}}$, $\mathbf{t}'_c$) → $\mathbf{out}_{\mathrm{bn}}$ | Ada_ResBlock: CONV(N256, K3x3), AdaIN, ReLU 
 Ada_ResBlock: CONV(N256, K3x3), AdaIN, ReLU 
 Ada_ResBlock: CONV(N256, K3x3,), AdaIN, ReLU 
 Ada_ResBlock: CONV(N256, K3x3), AdaIN, ReLU 
 Ada_ResBlock: CONV(N256, K3x3), AdaIN, ReLU 
 Ada_ResBlock: CONV(N256, K3x3), AdaIN, ReLU |
| **Up-sampling** | ($\mathbf{out}_{\mathrm{bn}}$, $\mathbf{t}'_c$) → $\mathbf{out}_{\mathrm{up}}$ | Ada_DECONV(N128, K4x4), AdaIN, ReLU 
 Ada_DECONV(N64, K4x4), AdaIN, ReLU 
 CONV(N1, K7x7) |
| **Output layer** | ($\mathbf{x}$, $\mathbf{out}_{\mathrm{up}}$) → $M_c(\mathbf{x})$ | $M_c(\mathbf{x}) = \tanh(\mathbf{x} + \mathbf{out}_{\mathrm{up}}) - \mathbf{x}$ |

Table 5: Attri-Net discriminator network architecture.

| Layers | Input → Output | Layer information |
|---|---|---|
| **Task embedding layer** | Task code $\mathbf{t}_c$ → Task embedding $\mathbf{t}'_c$ | $8 \times$ FC(L100,L100) |
| **Input layer** | | Ada_Conv: CONV(N64, K4x4), AdaIN, ReLU |
| **Hidden layers** | ($\mathbf{x}/\hat{\mathbf{x}}$, $\mathbf{t}'_c$) → $\mathbf{out}_{\mathrm{hid}}$ | Ada_Conv: CONV(N128, K4x4), AdaIN, ReLU 
 Ada_Conv: CONV(N256, K4x4), AdaIN, ReLU 
 Ada_Conv: CONV(N512, K4x4), AdaIN, ReLU 
 Ada_Conv: CONV(N1024, K4x4), AdaIN, ReLU 
 Ada_Conv: CONV(N2048, K4x4), AdaIN, ReLU |
| **Output layer** | $\mathbf{out}_{\mathrm{hid}}$ → $\mathcal{L}_{\mathrm{adv}}^{(c)}$ | CONV(N1, K3x3) |

