# OpenReview forum: "Inherently Interpretable Multi-Label Classification Using Class-Specific Counterfactuals"
_MIDL.io/2023/Conference — MIDL 2023 Poster_

### Official Review · Reviewer_qrAn · 2023-02-02

**Confidence:** 5
**Preliminary Rating:** 1

**Summary:**

The paper proposes a model architecture which is inherently interpretable. This is achieved using (which appears to be) a conditional autoencoder which generates a change to generate a counterfactual image. The method is applied to chest X-ray images.
................................................

**Strengths:**

The problem of explainable predictions is relevant and I agree that it is relevant in medical image analysis. The paper evaluates their approach on multiple datasets. The method is explained in detail and the overview figure is clear.

**Weaknesses:**

The framing of the paper does not seem correct. The method appears of limited novelty because it appears similar to the VA-GAN. The related work does not appear to be correct as it draws incorrect conclusions. Conclusions to be drawn from the experiments is unclear.

**Deanonymize Review:**

no

**Detailed Comments:**

I do not agree with the papers framing based on related work. I believe this derails the contribution of the paper. The paper concludes that "(CoDA-Nets) (Bohle et al., 2021) is, to our knowledge, the only existing model providing inherently interpretable visual explanations on a pixel-level." and then proceeds to only compare against this method as a counterfactual generation method. The paper cites VA-GANs (Baumgartner et al., 2018) and Gifsplanation (Cohen et al., 2021) which both generate counterfactuals. The paper even states that they use the VA-GAN and add a linear classifier to it.

The argument for what this paper is trying to achieve is not clear. Beyond the more complicated methods, the most basic comparison for an inherently interpretable model would be a model using attention over the input. The results of this method look very similar to the quality of Gifsplanation so it should be a baseline.

Other related work to look into:

> Using StyleGAN for Visual Interpretability of Deep Learning Models on Medical Images
> https://arxiv.org/abs/2101.07563
>
> xGEMs: Generating Examplars to Explain Black-Box Models
> https://arxiv.org/abs/1806.08867
>
> ExplainGAN: Model Explanation via Decision Boundary Crossing Transformations
> https://openaccess.thecvf.com/content_ECCV_2018/html/Nathan_Silberman_ExplainGAN_Model_Explanation_ECCV_2018_paper.html
>
> Explanation by Progressive Exaggeration
> https://arxiv.org/abs/1911.00483
>

It also should be argued why a model should be inherently interpretable instead of post hoc as this is a decision that the paper strives to achieve.

The experiments do not seem to be framed to prove or disprove anything hypothesis. Please state this clearly what can be concluded from the experiments.


**Paper Type:**

methodological development

**Questions To Address In The Rebuttal:**

Revisit the related work to better frame the contribution of this paper.

Make a clear set of claims of this work.

If the contribution is that this is a better inherently explainable method then compare with  inherently explainable methods like attention and one of the papers listed above. If this is a better counterfactual method then compare with counterfactual generation methods.

Frame the experiments to clearly support your claims.

---

### Official Review · Reviewer_scUd · 2023-02-04

**Confidence:** 4
**Preliminary Rating:** 4
**Recommendation:** Poster

**Summary:**

The paper proposes the use of a GAN to generate class-wise counterfactual maps for a given image such that these maps encode for the absence of a class when an additive function is applied to them. These attribution masks are then used to classify chest x-rays for a multi-label classification problem. The paper shows extensive experiments to highlight the at par performance of these new inherently interpretable models, while offering good interpretability characteristics like class sensitivity and improved explanatory power of the heatmaps as compared to previous methods.

**Strengths:**

1. The paper does a good job at motivating the problem and explaining the previous interpretability methods and tradeoffs associated with them.

2. The experimentation in the paper is strong and convincing. Multiple datasets, different ways of evaluating performance of the model, evaluating the interpretability (through class sensitivity scores), qualitatively evaluating the attribution heatmaps and other control experiments showing the impact of applying counterfactuals to negative samples, possibility of global interpretability, and ablations on loss terms.

3. The comparison of different interpretability methods on class sensitivity scores was interesting to see. Coda-Nets and SHAP coming out in the second and third place respectively is encouraging for inherently interpretable models like the one proposed in the paper.

**Weaknesses:**

1. The paper is missing some important previous work on inherently interpretable model and generation of counterfactuals for interpretability. The main omission is [1] which proposes a GAN-based method for generation of counterfactuals exactly like this paper. Hence the novelty of the idea is minimized. This paper uses the approach for medical imaging, does extensive experiments, and also introduces a few variations (GAN formulation, center loss, etc) and so is still useful for the community. But the related work section should include these papers and mention them appropriately. Some other missing references for inherent interpretability are [2, 3, 4].

2. The hypothesis that GAN-generated masks will generate relevant counterfactuals is dependent on the GAN objective and which distributions using the negative classes are used to align the generator as well as tricks like L1 regularization. There is a possibility that an incorrect counterfactual produce by the GAN model can still be useful for downstream prediction of the classes. As pointed in the previous reviews of [1], the interpretation of these counterfactuals is sometimes hard, making the attributions less useful. Note that this is a theoretical limitation, and the experiments do show the effectiveness of these trained attribution masks.

[1] Nemirovsky, D., Thiebaut, N., Xu, Y., & Gupta, A. (2020). Countergan: Generating realistic counterfactuals with residual generative adversarial nets. arXiv preprint arXiv:2009.05199.

[2] Cetin, I., Stephens, M., Camara, O., & Ballester, M. A. G. (2022). Attri-VAE: Attribute-based interpretable representations of medical images with variational autoencoders. Computerized Medical Imaging and Graphics, 102158.

[3] Javed, S. A., Juyal, D., Padigela, H., Taylor-Weiner, A., Yu, L., & Prakash, A. (2022). Additive MIL: Intrinsic Interpretability for Pathology. arXiv preprint arXiv:2206.01794.

[4] Agarwal, R., Melnick, L., Frosst, N., Zhang, X., Lengerich, B., Caruana, R., & Hinton, G. E. (2021). Neural additive models: Interpretable machine learning with neural nets. Advances in Neural Information Processing Systems, 34, 4699-4711.

**Deanonymize Review:**

no

**Paper Type:**

both

**Questions To Address In The Rebuttal:**

Would like to see the missing references be appropriately referenced in the paper along with a discussion of how this paper is different. The second point around the limitation of using a black-box generative model for generating counterfactuals also warrants a discussion. Specifically, what are the implications of using such counterfactuals for the downstream classifier? Is the risk of deploying a GAN on real-world data which is different than the training set the same as deploying a black box classifier with a post-hoc interpretability method used with it (generalization vs interpretability tradeoff)? Some discussion of potential limitations of this approach might also be a good idea.

---

### Official Review · Reviewer_gi8J · 2023-02-07

**Confidence:** 4
**Preliminary Rating:** 4

**Summary:**

In this study, the authors proposed an inherently interpretable visual explanation method Attri-Net specifically for the multi-label scenario. The model first generates class-specific attribution maps based on counterfactuals to identify which image regions correspond to certain medical findings. Then a simple logistic regression classifier is used to make predictions based solely on these counterfactual attribution maps. Comprehensive experiments were conducted on three widely used chest X-ray datasets, and the experiment results show that the Attri-Net produces high-quality inherently interpretable explanations with a high-class sensitivity while retaining classification performance comparable to state-of-the-art models.

**Strengths:**

In general, it is written. Comprehensive experiments were conducted on three widely used chest X-ray datasets, and the experiment results show that the Attri-Net produces high-quality inherently interpretable explanations with a high-class sensitivity while retaining classification performance comparable to state-of-the-art models.

**Weaknesses:**

1. In Eq(2) Why setting α0>α1?
2. The presentation of the algorithm part of this paper is confusing, and just Figure 1 can't show the proposed method well. The authors should reorganize the method section to facilitate understanding by researchers in related fields.
3. In Figure 5 of A.2. Global interpretability, the author said that "Attribution centers of disease and corresponding classifiers' weights provide a global explanation of Attri-Net". The global explanation of classifiers is not well understood here, because it is hard to really understand what the author means by the “global” explanation from the figure, i.e, what does the overall situation here refer to specifically? The authors should highlight the so-called “global” interpretation in this section or Figure 5.


**Deanonymize Review:**

no

**Paper Type:**

both

**Questions To Address In The Rebuttal:**

1. The presentation of the algorithm part of this paper is confusing, and just Figure 1 can't show the proposed method well. The authors should reorganize the method section to facilitate understanding by researchers in related fields.
2. In Figure 5 of A.2. Global interpretability, the author said that "Attribution centers of disease and corresponding classifiers' weights provide a global explanation of Attri-Net". The global explanation of classifiers is not well understood here, because it is hard to really understand what the author means by the “global” explanation from the figure, i.e, what does the overall situation here refer to specifically? The authors should highlight the so-called “global” interpretation in this section or Figure 5.

---

### Meta-Review · Area_Chair_TcRK · 2023-02-22

**Recommendation:** Accept (Poster)
**Confidence:** 3

**Metareview:**

The paper proposes a model that is inherently interpretable, based on counterfactuals, and is suitable for a multi-label scenario, here applied on chest x-rays. The reviewers are somewhat divided in their scores. Two of three reviewers raise concerns about the framing of the paper / its contributions vs other literature (although the reference list is quite extensive for a conference paper). The authors seemed to have clarified some of these points in their revised paper, although not all reviewers responded with revised comments.

Although not all the reviewers responded in follow-up, it seems the topic is quite current and worthy of broader discussion in the community. With that thought I would recommend acceptance, though it is a borderline case so the final decision might depend on the ranking vs other papers.